# Research Progress on MET, Tip Link, and Stereocilia Complex with Special Reference to Zebrafish

**DOI:** 10.3390/ijms26178480

**Published:** 2025-08-31

**Authors:** Xun Wang, Yuqian Shen, Dong Liu

**Affiliations:** Nantong Laboratory of Development and Diseases, School of Life Sciences, Co-Innovation Center of Neuroregeneration, Nantong University, Nantong 226001, China; shuin@stmail.ntu.edu.cn

**Keywords:** sensory hair cells, stereocilia bundles, mechano-electrical transduction, tip link, zebrafish

## Abstract

Hearing is essential for animal survival and social communication, relying on the function of sensory hair cells. These cells possess organized stereocilia bundles enriched with mechano-electrical transduction (MET) channels that convert mechanical stimuli into electrical signals. Tip links, fine extracellular filaments connecting adjacent stereocilia, play a critical role in transmitting mechanical forces to MET channels. Over the past three decades, technological advances have significantly enhanced our understanding of the molecular and cellular mechanisms underlying auditory transduction. Zebrafish, with its conserved hair cell structure and function similar to mammals, has become a valuable model in auditory research. The aim of this review is to summarize the research progress on the molecular and cellular mechanisms of MET, tip link, and stereocilia complex, with an emphasis on zebrafish studies, providing an important reference for understanding diseases of the human auditory system.

## 1. Introduction

Hearing is a vital sensory ability that is essential for human and animal survival, communication, and interaction with their environment. In mammals, it enables the perception and processing of acoustic signals, thereby facilitating communication, spatial orientation, and environmental awareness [1]. Anatomically, the mammalian ear is structured into the outer, middle, and inner ear, with the inner ear housing the critical sensory components responsible for sound detection and balance regulation (Figure 1). For animals, hearing is essential not only for intraspecies communication and predator avoidance but also for behaviors crucial to reproduction and foraging. In humans, it further supports the formation of social relationships and cultural expression through language and music.

Disruptions in auditory function are associated with a range of disorders, including sensorineural hearing loss, vestibular dysfunction, tinnitus, and hereditary hearing impairments. These conditions are often rooted in defects in the structure or molecular machinery of the auditory system. For instance, damage to inner ear hair cells (IHCs) or the auditory nerve can result in sensorineural deafness [2], while dysfunction in vestibular hair cells or neurons can impair balance and spatial orientation [3]. Tinnitus, although complex in origin, has been linked to abnormalities in both peripheral and central auditory pathways [4,5,6].

At the core of auditory signal transduction are the IHCs-specialized epithelial cells with precisely organized stereocilia bundles that convert mechanical stimuli into electrical signals [1,7]. In the inner ear, hair cells are predominantly found in the Corti’s apparatus (Corti), the vestibular organ, and the semicircular canals. Hair cells in the Corti are classified into IHCs and outer hair cells (OHCs) based on their location, function, and cilia, and they are primarily responsible for transmitting auditory information (Figure 2a) [8]. Hair cells in the vestibular apparatus and semicircular canals are classified into two types: type I and type II hair cells (Figure 2b). These hair cells primarily function in detecting head movements, transmitting positional information, and maintaining body balance [9].

Hair cells in the auditory system are primarily composed of stereocilia, MET channels, and tip links. Detailed stereocilia architecture in a zebrafish hair cell, showing the three distinct rows and the localization of MET channels near the tips of shorter row stereocilia. Tip links connecting adjacent stereocilia are also depicted (Figure 3). Stereocilia are organized into a staircase-like, highly ordered arrangement on the apical surface of hair cells. Tip links, delicate extracellular filaments, connect the tips of shorter stereocilia to their taller neighbors and are essential for mechanotransduction. When sound-induced movement of endolymph causes the stereocilia to deflect, tension on the tip links opens MET channels located at the tips of the shorter stereocilia, allowing the influx of ions and initiating the electrical signal. The elucidation of the molecular composition of tip links began with the identification of mutations in *CDH23* and *PCDH15*. Mutations in *CDH23* were found to cause Usher syndrome type 1D (USH1D) and nonsyndromic deafness DFNB12 [10,11], while mutations in *PCDH15* were associated with Usher syndrome type 1F (USH1F) and nonsyndromic deafness DFNB23 [12]. These discoveries established cadherin-23 and protocadherin-15 as core structural components of the stereociliary tip link in hair cells.

MET channels are complex multiprotein structures critical for auditory signal transduction. In 2002, positional cloning following linkage analysis of families with autosomal recessive nonsyndromic deafness (DFNB7/11) led to the discovery of pathogenic mutations in *TMC1*, causing hearing loss [13]. This milestone provided the first genetic evidence linking TMC1 to auditory mechanotransduction. In 2018, research identified Transmembrane Channel Protein 1 (TMC1) as an essential component of the MET channel. TMC1 possesses 10 transmembrane structural domains, with four of which are localized to the MET channel [14]. A recent study pointed out that *TMC1* in *C. elegans* can adapt to various sensory functions, such as touch and base sensation, with the help of different auxiliary proteins, and the specific structural domains of TMC-1 allow it to fulfill these diverse functions [15]. Similarly, using *C. elegans* as a model, researchers have described the single-particle cryo-electron microscopy (cryo-EM) structure of the native TMC1 complex, revealing a dimeric assembly accompanied by auxiliary subunits CALM-1 and TMIE [16]. This finding is of great significance for understanding the functions and regulatory mechanisms of the auditory system in hair cells, and provides new clues and breakthroughs for the study of auditory perception.

In the vestibular system, additional components are involved in MET channel assembly and regulation. Research showed that CIB2 and CIB3 play important roles in maintaining the function of MET channels in vestibular hair cells. While knockdown of either CIB2 or CIB3 alone did not significantly impair electrical signaling, simultaneous loss of both proteins resulted in severe vestibular dysfunction in mice [17]. This suggests that CIB2 and CIB3 exhibit complementary transcriptional expression patterns and exert region-specific, redundant functions. These findings provide important insights into the molecular mechanisms underlying vestibular function. Other essential MET-associated proteins include TMIE (Transmembrane Inner Ear Protein) and LHFPL5 (LHFPL tetraspan membrane protein 5) [18]. TMIE functions mainly in the inner ear’s tonotropic cells and enhances MET receptor signaling in the MET complex, indirectly affecting MET signaling activation and transmission by promoting the membrane localization and stability of the receptor, thus playing a crucial role in cellular function and response. The mechanism may involve direct interaction with the MET receptor or modulation of signaling pathways downstream of the receptor [19]. Recent studies show that LHFPL5, a tetraspan four-transmembrane protein, stabilizes the MET receptor complex by interacting with PCDH15, TMC1/2, and TMIE, regulates its trafficking and internalization, and thereby influences downstream cellular behaviors such as proliferation, migration, and hair bundle morphogenesis [20].

Together, these components illustrate the intricate protein network that underlies MET channel formation and function in hair cells. Understanding their roles not only advances our knowledge of auditory transduction but also provides potential targets for therapeutic intervention in hearing and balance disorders.

## 2. Zebrafish Is an Important Model Organism for Studying Hair Cell Function

A variety of model organisms are widely used in the field of auditory research, including mice, zebrafish, *Drosophila*, and *Caenorhabditis* elegans (*C. elegans*). Each of these model organisms has unique strengths, but they also have some limitations, particularly in terms of their evolutionary background, regenerative capacity, and number of hair cells. For example, mice, as mammalian models, are important in auditory studies because they are evolutionarily closer to humans. However, the auditory system of mice differs from that of humans in several aspects, including the range of auditory frequencies. In addition, the evolutionary properties of the mouse auditory system may not be fully applicable to explain the auditory properties of other mammals, especially humans [21]. Moreover, despite extensive efforts, the complete molecular composition of the mechanoelectrical transduction (MET) channel remains unclear. In mammals, the limited number of hair cells presents a technical challenge for dissecting MET components. *Drosophila* is a valuable genetic model for studying auditory mechanosensation, enabling the discovery of channels like *NOMPC* [22] and Piezo [23] and linking hearing to behavior via courtship song [24]. However, it lacks vertebrate-like hair cells, limiting its relevance to studies of hair cell function and human hearing loss. Although *C. elegans* has been used to resolve the structure of TMC1 [16], it lacks hair cells, which restricts its utility for comprehensive MET studies. In contrast, zebrafish possess a large number of accessible hair cells, making them a powerful model for elucidating the key components and structure of the MET channel.

Zebrafish possess a series of unique advantages that can overcome the limitations of in vitro experiments and other model animals, and these advantages make zebrafish one of the favored model organisms for scientific research. Firstly, zebrafish possess a high reproductive output. This means that researchers can easily obtain a large number of experimental samples for large-scale experimental studies. Secondly, the genetic manipulation in zebrafish is relatively straightforward, allowing researchers to easily perform genetic studies using techniques such as CRISPR/Cas9, which enables the precise investigation of specific genes related to auditory function [25]. Additionally, zebrafish embryos are completely transparent, enabling researchers to easily observe and document their development, including the development of the auditory system. Finally, zebrafish are well-suited for live imaging, allowing real-time observations and experiments on living animals, which provides valuable insights into their biological functions.

Although zebrafish are evolutionarily distant from humans, the fundamental structure and function of their inner ear are similar to those of the human auditory system, making them a valuable model for studying the mechanisms underlying hearing [26,27]. Zebrafish possess many physiological systems, including sensory organs, that are conserved with those of mammals. Although zebrafish lack a cochlea-like structure in their inner ear, components such as the saccule and utricle perform analogous functions: the saccule is involved in auditory perception [28], while the utricle plays a critical role in balance and spatial orientation [29]. Hair cells in the zebrafish inner ear are morphologically and functionally similar to mammalian IHCs and serve as the primary sensors for hearing and equilibrium. In the zebrafish otic vesicle, crista hair cells are further classified into anterior crista hair cells (ACHC), lateral crista hair cells (LCHC), and posterior crista hair cells (PCHC). Zebrafish possess lateral line hair cells located on the body surface, which are absent in mammals (Figure 4). The lateral line system is a specialized mechanosensory organ unique to aquatic vertebrates, enabling them to detect water movement, temperature changes, and nearby objects. This system plays a crucial role in predator avoidance, prey detection, and navigation, offering a complementary model for studying mechanosensation. Compared to mammals, the inner ear and lateral line system of zebrafish share a similar hair cell structure, including the MET channel, the tip link, and cilium complexes. Studies have shown that zebrafish MET channels play a crucial role in hearing and balance, with tip link proteins attached to the tips of cilia. These proteins open the MET channels through mechanical tension, allowing the influx of ions and generating electrical signals [30]. Tip link proteins are mainly composed of *Cdh23* and *Pcdh15* and form transmembrane connections between cilia [26]. Zebrafish cilia share a similar structure to humans, making them an ideal model for studying hair cell function. Zebrafish *tmc1* and *tmc2a* have been shown to interact with distinct cytoplasmic domain isoforms of *pcdh15a*, referred to as CD1 and CD3, which differ in their C-terminal sequences. These isoforms differentially affect the stability and assembly of the mechanotransduction complex within stereocilia [31]. This structural and functional conservation supports the use of zebrafish as an effective model for investigating human auditory function.

Additionally, zebrafish possess several unique advantages in the study of auditory function. One notable feature is the remarkable self-repair and regenerative capacity of their hair cells. Single-cell transcriptomic analyses have identified molecular differences among sensory hair cell and supporting cell subtypes in the zebrafish inner ear [32]. Furthermore, Single-cell transcriptome analysis revealed three sequential phases of gene expression during the regeneration of zebrafish sensory hair cells [33]. This feature makes zebrafish an excellent system for studying the recovery mechanism of hearing damage.

However, despite these strengths, there are notable limitations of the zebrafish model in auditory research. Its auditory system is anatomically simpler than that of mammals, lacking structures such as a cochlea and certain specialized auditory organs, which may constrain its ability to fully recapitulate aspects of mammalian hearing. Functional assays for auditory perception in zebrafish remain challenging, with behavioral readouts sometimes limited by environmental variables such as water quality and temperature. Additionally, current imaging techniques face difficulties in visualizing deep inner ear structures at high resolution. These constraints necessitate cautious interpretation when extrapolating findings from zebrafish to mammals.

To better illustrate these points, we provide a comprehensive visual comparison of zebrafish, mice, *Drosophila*, and *C. elegans* as model organisms in auditory and mechanosensory research. The figure integrates key anatomical features, genetic tools availability, and experimental advantages and limitations for each model (Figure 5). This comparative overview facilitates a clearer understanding of the strengths and constraints inherent to each system, highlighting why zebrafish serve as a valuable yet imperfect intermediary model between invertebrates and mammals.

Overall, zebrafish offer distinct advantages, including ease of genetic manipulation, a strong capacity for hair cell regeneration, and transparent developmental stages that facilitate in vivo imaging and analysis. Importantly, zebrafish share many key features with mammals, including similar sensory hair cell morphology, tip link and stereocilia complexes, and auditory signaling pathways. These attributes make zebrafish a powerful model organism for studying auditory biology and the molecular basis of hearing loss, contributing significantly to our understanding of auditory function and the development of potential therapeutic strategies.

## 3. Research Progress on MET Channels in Zebrafish

In mammals, MET channels play a key role in the auditory system. These channels are located near the tips of the shorter row stereocilia and convert sound-induced mechanical stimuli into electrical signals. The *TMC* gene family encodes transmembrane channel proteins (e.g., TMC1 and TMC2) that are the essential components of MET channels and play a critical role in mammalian auditory transduction. However, the precise mechanisms of TMC1 and TMC2 in channel function and their regulation are not yet fully understood [13]. For example, there is still a lack of sufficient evidence on how to regulate the opening and closing of these channels and their interactions with other proteins [34]. Current studies have identified the molecular composition and structure of MET channels, which consist of specific protein subunits, such as TRPML3, TRPV4, TRPA1, and PIEZO2 in the TRP family. Additionally, CIB2 and CIB3 form complexes with TMC1 and TMC2, which play a role in auditory mechanotransduction. However, much remains unknown about the specific regulatory mechanisms and functions of these complexes [35]. Despite significant progress in our understanding of MET channels, some key questions remain unanswered, such as how to precisely control their opening and closing and how they adapt in different sensory modalities. The molecular composition of MET channels in zebrafish is the focus of research. It was noted that Tmie is essential for localizing Tmc1 and Tmc2b in zebrafish sensory hair cells. Without Tmie, these MET channel subunits cannot be correctly localized, resulting in impaired mechanotransduction function [36]. Previously, it was found that different combinations of *TMC1/2a/2b* genes are essential for proper function in different subtypes of zebrafish inner ear hair cells [25]. In zebrafish, two isoforms of *lhfpl5*, *lhfpl5a* and *lhfpl5b*, have been shown to be essential for the mechanotransduction of different types of sensory hair cells [37]. By analyzing the molecular composition of MET channels in zebrafish, their structure and function can be better understood. A recent study reported that *esrp1* and *esrp2* regulate the expression and function of MET channels by stabilizing the mRNA levels of zebrafish *tmc1* and *tmc2a* [38]. This finding provides the first evidence that MET channel activity is subject to post-transcriptional regulation, complementing the current understanding centered on gene coding and membrane localization mechanisms.

Zebrafish as a model organism provides important complementary evidence in studying the function of core molecules of the MET channel. Through gene editing techniques, the researchers discovered that mutations in the *tmc1*, *tmc2a*, and *tmc2b* genes result in functional abnormalities in zebrafish inner ear hair cells, indicating that different *Tmc* isoforms play distinct roles in these cells. The zebrafish study not only bridges the gap in mammalian studies but also provides new perspectives and experimental tools to gain insight into the function and regulatory mechanisms of MET channels. For example, it was found that *TMC1* mutations lead to the loss of inner ear hair cells, while *TMC2* mutations affect the function of different subtypes of inner ear hair cells [39]. This suggests that the roles of TMC1 and TMC2 in inner ear hair cells may be more complex than previously thought. With the zebrafish model, researchers were able to observe the roles and interrelationships of these genes in detail to better understand their specific functions in auditory conduction.

The formation and functioning of MET channels during zebrafish inner ear development are important for understanding the development of the auditory system and the etiology of related diseases. It has been shown that ammonia adversely affects the lateral line sensory organs of the zebrafish embryo, including damage to hair cells and impaired mechanotransduction functions [40]. Aminoglycoside-induced hearing loss stems from damage or loss of mechanosensory hair cells in the inner ear. Modification of SS-31 peptide significantly reduced the activity of MET channels and gentamicin uptake in zebrafish lateral line hair cells, suggesting that SS-31 may reduce the damage of aminoglycosides to the auditory system by localizing the drug to the mitochondria [41]. These models provide an important tool for studying the function of MET channels and the mechanisms associated with auditory disorders. Recent studies have shown that CIB2 and CIB3 form stable structural complexes with TMC1/2. AlphaFold2 modeling and molecular dynamics simulations revealed their critical roles in channel conformation and ion conduction [42]. Zebrafish experiments further confirmed their physiological relevance, marking a significant advance in structural understanding of MET channels.

Using zebrafish as a drug screening platform, drugs targeting MET channels can be identified and evaluated, such as Mavacamten, a drug targeting heart disease that acts on mechano-electrical transduction channels in muscle. Its role in MET channels has not been studied and can be further explored using zebrafish. Effective strategies and tools for therapeutic drug development for hearing impairment [43]. A recent study developed a pharmacophore-based screening pipeline, identifying and validating dozens of compounds from a 22-million compound library that modulate *TMC1* activity [44]. The zebrafish model facilitates in vivo functional assays using behavioral and fluorescence-based readouts in multi-well formats, allowing rapid and cost-effective evaluation of large compound libraries [45,46]. These candidate molecules exhibit MET channel-blocking and otoprotective potential, offering new avenues for MET-targeted therapies. Through such approaches, zebrafish models help deepen understanding of MET channel structure, function, and associated auditory disease mechanisms.

## 4. Advances in MET Research by Single-Cell Sequencing

Single-cell sequencing technology has significantly advanced the study of MET, particularly by elucidating the functions and mechanisms of MET channels across different cell types. Single-cell transcriptome sequencing has revealed gene expression changes in these cells under mechanical stress, resolving mechanosensitive channels in functional and pathological states of the heart [47]. Analyses of taste and somatosensory neurons in the mouse geniculate ganglion identified the transcriptome and neurotransmitter response profiles of these neurons and identified novel markers, Phox2b and Prrxl1, which can help to differentiate between different types of neurons [48]. Using single-cell sequencing, researchers have found that the frequency of calcium activity mediated by the mechanosensitive channel Piezo1 in the apical cell branches of cerebrovascular endothelium determines the fate of apical cell branch contraction or elongation, which in turn determines the choice of pathways for blood vessel growth and the pattern of formation of the three-dimensional network of cerebrovascular vessels [49]. On the disease-related side, single-cell and large-scale RNA sequencing of renal cells from diabetic mice (BTBR *ob*/*ob*) revealed a glucose-independent response in glomerular cell types in early diabetic nephropathy. The upstream gene regulatory network revealed that the mechanosensitive transcriptional pathway MRTF-SRF is predominantly activated in thylakoid cells [50]. Using Patch-seq experiments on mouse dorsal root ganglion (DRG) neurons, combined with mechanosensitive current recordings and single-cell RNA sequencing, it was found that the candidate gene *Piezo2*, which is associated with specific mechanosensory functions, is enriched in rapidly adapting mechanosensitive current-expressing neurons, whereas *Tmem120a* and *Tmem150c* are expressed uniformly in all mechanosensory neuron subtypes are uniformly expressed in all mechanosensory neuron subtypes [51]. Knockdown experiments further confirmed that Tmem120a and Tmem150c proteins do not mediate mechanosensitive currents. This dataset provides a cell-type-specific open resource for exploring mechanosensory properties.

The core function of the auditory system is to convert mechanical energy into electrical signals, primarily achieved through MET channels on hair cells. The study of MET channels is of great significance for understanding auditory function and treating hearing disorders. However, the limited number of hair cells and their complex microstructure make it challenging to elucidate their fine molecular mechanisms using traditional bulk sequencing methods. The development of single-cell sequencing technology has provided us with a more precise means of research, leading to breakthroughs in the study of hair cells and their MET channels. Single-cell sequencing technology enables high-resolution analysis of the genome, transcriptome, and epigenome at the level of individual cells. In particular, single-cell transcriptome sequencing technology has revealed the heterogeneity of hair cells, helping researchers better understand the functional characteristics of different types of hair cells. For example, Burns and colleagues used single-cell RNA sequencing to identify specific gene expression profiles of various hair cell types in the inner ear, highlighting the utility of this technology in advancing our understanding of auditory cell specialization and function [52]. Through single-cell transcriptomic analysis, the researchers identified 14 sensory and non-sensory cell subtypes in the otolithic apparatus of neonatal mice [53].

Studies on MET channel-related genes in single-cell sequencing have found that *Lhfpl5* (also known as TMHS, DFNB66/67), which is essential for HC mechanotransduction, is progressively increased in all SGNs (Spiral Ganglion Neurons) during differentiation, whereas *Myo6* (DFNA22/DFNB37), which is essential for hair cell structural integrity, is also expressed in all SGNs at early stages [54]. By comparing single-cell transcriptome profiles before and after gene therapy interventions, we can assess the efficacy of gene therapy at single-cell resolution, thus helping to optimize future gene therapy strategies. Iwasa and colleagues’ use of the Woodchuck Hepatitis Virus post-transcriptional regulatory element (WPRE) to augment Tmc1 transgene expression resulted in poor hearing recovery [55], and found that Tmc1 expression in OHCs was higher in animals treated with WPRE-containing vectors than in the group without WPRE by scRNA-seq analysis, with a relatively better therapeutic efficacy, suggesting that optimizing the dose of the transgene expressed in the target cells is essential for gene therapy for hearing loss.

GFP^+^ cells isolated from transgenic zebrafish larvae, all of which express green fluorescent protein GFP in their hair cells, were analyzed using scRNA-seq. Three subtypes of hair cells (i.e., macula hair cell (MHC), crista hair cell (CHC), and neuromast hair cell (NHC)) were characterized and validated by whole-tissue in situ hybridization analysis of marker genes [56]. The distribution of cells expressing genes encoding MET and tip link components was indicated, and it has been previously described that MET channels are required for functional hair cells, which are a complex consisting of several components such as TMC1, TMC2, TMIE, LHFPL5, and CIB2. Hair cells with functional MET channels are essential for normal hearing and balance. In addition, tip junctions, which play an important role in cilia deflection and MET channel gating, are composed of two calcineurin proteins, CDH23 and PCDH15. To determine whether these key molecules are also expressed in zebrafish hair cells and to understand how they differ between hair cells, the team analyzed the expression patterns of the genes encoding these important proteins. Genes homologous to components of the mammalian MET complex, such as *tmc1*, *tmc2a*, *tmc2b*, *lhfpl5a*, *tmie*, and *cib2,* are all expressed in NHCs, MHCs, and CHCs in zebrafish. However, *tmc2b* shows lower expression in MHCs, and *lhfpl5b* exhibits reduced expression across all three hair cell types. Additionally, the genes encoding tip link components, *cdh23* and *pcdh15a*, are expressed in a limited subset of hair cells and are also detected in some supporting cells. Moreover, several other genes were found to be highly expressed in hair cells, although their specific roles in hair cell function have not yet been characterized.

The application of single-cell sequencing technology to MET channel studies has greatly advanced the understanding of hair cell function and auditory mechanisms. By identifying specific genes, resolving gene regulatory networks, and investigating disease mechanisms, single-cell sequencing technology provides an important basis for developing new treatments for hearing disorders. In the future, with the further development of the technology, single-cell sequencing will play an even more important role in the study of the auditory system.

## 5. Mutations of MET Associated with the Auditory System in Zebrafish

Genetic mutations are a major cause of hearing loss, and mutations in MET channels have been linked to hearing loss. A study combined behavioral and genetic analyses to investigate the mechanoreceptive function of zebrafish hair cells. The results showed that mutations in multiple genes interfere with the function of MET channels, thus affecting the behavioral responses of zebrafish [57]. In recent years, zebrafish have been widely used to study the molecular mechanisms underlying auditory and balance functions, particularly how mutations in genes related to MET channels impact auditory and balance functions in zebrafish.

Firstly, genetic mutations affect MET channel structure and function. Mutations in the *cdh23* gene in zebrafish have been found in previous studies to affect the apical linkage structure of hair cells, which in turn affects the function of mechanoelectrical transduction channels [26]. The function of the *pcdh15* gene in zebrafish has been found to be associated with auditory and balance functions [58]. PCDH15 interacts physically and functionally with TMC1 and TMC2, and PCDH15 may play a role in anchoring TMC proteins to the top chain or facilitating their integration into the mechanotransduction machinery [31]. In addition, studies in zebrafish have explored how mutations in kinocilia-associated genes affect hair cell mechanosensitivity [59]. For example, deletion of *grxcr1* has been shown to impair the mechanoelectrical transduction function of hair cells, leading to hearing loss or hearing impairment [60]. These findings highlight the critical role of MET channels in the auditory system and provide valuable insights into the link between genetic mutations and auditory functions [61].

Secondly, some genetic mutations are closely associated with the development of auditory disorders. Whitfield’s study pointed out that several zebrafish mutant models associated with auditory disorders, demonstrating that these mutations can directly impact the structure or function of the MET channel, ultimately leading to auditory dysfunction [62]. Additionally, zebrafish models of mechanoreceptive hair cell dysfunction in Usher syndrome 3 reveal Clar1 as an essential hair cell bundle protein [63]. Mutations in the *loxhd1* gene have also been shown to disrupt or eliminate LOXHD1 protein function, thereby impairing mechanotransduction in inner ear hair cells [64]. These findings emphasize the importance of genetic mutations in the auditory system and their contribution to hearing loss.

In addition, mutations may affect the regulatory mechanisms of MET channels, including their expression, localization, and interactions with other proteins in auditory cells, thereby affecting the proper functioning of auditory functions. The synaptic ribbon is an electron-dense structure tying synaptic vesicles to the presynaptic region of mechanosensory hair cells adjacent to the postsynaptic terminals of afferent fibers. This has been investigated by looking at mutants lacking innervation of hair cells and observing the effect of postsynaptic elements on ribbon formation and maintenance in the zebrafish lateral line system was investigated in mutants lacking hair cell innervation. The results showed that initial ribbon formation does not require innervation [65]. Sensory hair cells are key cells in the inner ear and lateral line system responsible for converting mechanical stimuli into neural signals for auditory and balance functions. These observations from the lateral line provide important insights into auditory hair cell polarity and mechanotransduction mechanisms, highlighting conserved principles across sensory systems. It has been shown that the *circler* gene mutants in sensory hair cells affect the normal function of mechanoreception. Specifically, these mutants show morphological and functional defects in sensory hair cells that lead to abnormalities in hearing and balance [66]. Thus, by using gene mutation models for auditory drug screening, researchers can assess the effects of different drugs on MET channel function and thus discover new auditory therapeutic agents. Finally, by studying gene mutation models, it is possible to reveal the regulatory mechanisms of MET channels during the development of the auditory organ, including their role in hair cell formation, differentiation, and synaptic connections [59]. Specifically, some genetic mutations may lead to abnormalities in MET channel function, affecting mechanoreception and signaling in auditory cells [26,31,63].

These studies are of great value to the study of human deafness diseases, revealing the relationship between gene mutations and auditory dysfunction, and providing new ideas and methods for the diagnosis and treatment of deafness.

## 6. Advances in the Study of Tip Links in Zebrafish

Tip link proteins have multiple important functions in auditory cells, including connecting cilia, maintaining the stability of ciliary structures, converting mechanical energy into neural signals, and regulating the development of the auditory system. These tip links are mainly composed of CDH23 and PCDH15, which form transmembrane connections between cilia, ensuring the ability of auditory cells to perceive mechanical stimuli. While earlier studies provided foundational insights into the structure and function of tip links, more recent research has uncovered their complex and dynamic properties [67,68,69]. The *Cdh23* gene encodes a cadherin protein that serves as an essential component of the inner ear hair cell tip link. Critical for transduction channels that transmit mechanically stimulated signals to hair cells, it has been shown that splicing abnormalities in exon 68 of the *Cdh23* gene result in impaired stability of the parietal chain, which in turn affects the transmission of mechanically stimulated signals and the function of inner ear hair cells. This loss of stability leads to a gradual decline in hearing and ultimately to hearing loss [70]. PCDH15 is a key component of the hair cell parietal chain and is essential for the transmission of mechanically stimulated signals to the transduction channels inside the hair cell. Studies have analyzed the three-dimensional structure of PCDH15 and its complexes on hair cells using cryo-electron tomography in order to obtain the true structure of PCDH15 in hair cells, and have observed its distribution and arrangement on sensory filaments [71]. Furthermore, by comparing the nanomechanical properties of wild-type and mutant PCDH15 dimers, researchers can gain insight into how mutations affect the mechanical properties of the top chain and further reveal the relationship between the structure and function of the top chain [72]. This connective structure maintains the stability of the cilia, allowing auditory cells to accurately perceive mechanical stimuli from the external environment, and the researchers used single-molecule force spectroscopy- a technique that involves investigating the mechanical properties of hair cell tip links by stretching a single molecule and measuring the force they exert during the deformation. Through this method, they were able to quantify the strength and stability of the tip link when subjected to forces, providing insights into their dynamic behavior under different force conditions [73]. Researchers have also used nanomechanical techniques, such as atomic force microscopy (AFM), to study the behavior of the tip link’s calcineurin when subjected to forces [74].

In addition, tip link proteins play an important role in the functional regulation of the auditory system. Tip link proteins are involved in the adaptive regulation of the auditory system in response to the external environment, and their regulatory mechanisms may involve the regulation of multiple signaling channels [75]. Recent studies have prepared PVDF nanofibres using electrospinning technology and applied them to artificial hair cell structures as the connecting tip part. They investigated the piezoelectric properties of PVDF nanofibres under mechanical stimulation, as well as their stability and reliability in artificial hair cell structures [76]. Therefore, tip link proteins could be a potential target for future therapeutic treatments of auditory disorders, and the related clinical treatments could be one of the focuses of research [77].

In summary, tip link proteins play multiple roles in auditory cells, including maintaining the stability of cilia structure, converting mechanical energy into neural signals, regulating the development and function of the auditory system, as well as the relationship with the occurrence of auditory diseases and drug therapy. The in-depth study of tip link proteins not only helps us to better understand the operation of the auditory system, but also provides new ideas and possibilities for the diagnosis and treatment of related auditory diseases.

In zebrafish, tip link proteins are important components of the auditory system, connecting filaments of auditory hair cells and playing a key role in sound signal transmission and auditory perception. Caberlotto and colleagues pointed out the compositional and structural features of tip link proteins and highlighted their mechanosensitivities in the zebrafish auditory system through a study [78]. Zebrafish Cdh23 proteins are concentrated near the tips of hair bundles and are essential tip link components required for functional transduction of hair cells [26]. Further studies have found that aberrant expression or loss of function of tip link proteins leads to impaired auditory function in zebrafish, which adversely affects both survival and behavior. Research has shown that the regulatory mechanism of tip link proteins plays a key role in the regulation of auditory adaptations in zebrafish, including their mechanosensitivity in auditory perception and their interaction with other molecules [22,31]. In particular, tip link proteins such as *Pcdh15* interact with other components of the mechanotransduction machinery, forming an essential part of the molecular complex responsible for converting mechanical stimuli into electrical signals in auditory hair cells [30]. Previous studies have also shown that the tip link calcineurin *Pcdh15a*, *Cdh23*, and the *Myo7aa* motor protein are required for the correct positioning of the stereocilia tip to *Lhfpl5a* [37].

In addition to contributing to the development and function of the auditory system, the study of tip link proteins has also provided insights into the development of therapeutic approaches for hearing impairments. The study further revealed the regulatory mechanism of tip link proteins, providing an important research basis for the development of novel auditory therapeutics [79]. However, research on the impact of aberrant tip link protein expression on auditory function in zebrafish remains limited.

Moreover, the study of tip link proteins has provided insights into the engineering applications of the auditory system. Research demonstrated that the structure-function relationship of tip link proteins elucidates the underlying mechanisms of auditory perception and offers important insights for the design and development of cochlear implants [80]. Zebrafish possess a strong ability to regenerate hair cells and restore function after damage. However, direct evidence showing that regeneration can fully recover defects caused by specific MET or tip-link gene mutations, which lead to permanent deafness in mammals, remains limited. Recent studies indicate that stem and progenitor cell proliferation in zebrafish inner ear is regulated by cell–type–specific *cyclin D* genes, highlighting molecular pathways that control regeneration [81]. Nonetheless, zebrafish offer a valuable model to study potential regenerative therapies for genetic hearing loss [82]. A recent study found that the Tip links may exhibit a slip-ideal-slip behavior under mechanical force. Specifically, they may undergo minor displacements under low levels of force, followed by larger shifts only after a certain force threshold is exceeded. This mechanical response allows tip links to function as a filter during sound transmission—only when the auditory force surpasses a specific threshold will it activate the sensory cells. This mechanism enhances both the sensitivity and selectivity of the auditory system to sound stimuli [83].

In summary, the zebrafish model offers unique advantages and research value. As a model organism, zebrafish has an auditory system and hair cell structure similar to that of humans and is highly genetically manipulable, making it an ideal tool for studying the function and mechanism of tip link proteins [84]. With the zebrafish model, researchers have been able to perform genetic mutation experiments to see how these mutations affect the structure and function of tip link proteins, and in turn, how these changes can lead to hearing deficits. This approach complements mammalian models by providing more detailed observations and experimental tools, advancing a deeper understanding of the role of tip link proteins in the auditory system [85,86].

## 7. Mutations of Tip Link Associated with the Auditory System in Zebrafish

Over the past few years, it has been found that the zebrafish tip link proteins, a group of protein connections between auditory hair cells that transmit mechanical stimuli to aid in the perception of sound, are strongly associated with hearing-related diseases. By studying zebrafish models, scientists have found that abnormal expression or loss of function of tip link proteins is strongly associated with inherited hearing disorders. Through these studies, we can not only understand the function of zebrafish tip link proteins, but also apply these findings to the study of human hearing disorders [87].

Firstly, the normal function of MYO6 helps to maintain the cell membrane stability of hair cells and the correct configuration of tip links, which affects auditory and vestibular function, and zebrafish models show that *myo6* mutations lead to severe ear defects [88]. This supports a strong relationship between tip link-associated proteins and auditory function. For example, Research showed that deletion of the *cdh23* gene, which encodes a tip link-associated protein, impairs auditory hair cell function and leads to hearing loss in zebrafish [26]. These findings underscore the essential role of tip link proteins in maintaining the structure and function of auditory hair cells. Moreover, tip link proteins interact with other key components in the auditory sensory pathway to form an integral part of mechanotransduction. This suggests that tip link proteins play a key role in maintaining the structure and function of auditory hair cells. Moreover, tip link proteins interact with other key components in the auditory sensory pathway to form an integral part of mechanotransduction. Notably, Research elucidated the interaction between Cdh23 and Pcdh15 proteins, both of which are essential components of auditory hair cells [89], further emphasizing the crucial role of the tip link complex in the auditory system.

The association of tip link proteins with hearing-related diseases has also been gradually revealed. Analysis of the human gene mutation database indicates that mutations in tip link proteins are associated with diseases such as congenital deafness [31]. In zebrafish models, this association can be further explored using gene knockout technology targeting tip link-related genes, which provide direct insights into the molecular mechanisms underlying hearing impairments [90]. Several studies have demonstrated that dysfunction of tip link proteins in zebrafish is indeed correlated with hearing-related disorders [81]. Furthermore, it has been shown that age-related weakening of the tip link protein interaction network contributes to progressive hearing loss [86]. These findings highlight the importance of tip link proteins in auditory function and provide new ideas and methods for the diagnosis and treatment of human hearing-related diseases.

## 8. Advances in the Study of the Cilium Complex in Zebrafish

Cilia are important structures in inner ear sensory hair cells and are responsible for converting mechanical stimuli into neural signals. RGS12 was found to polarize the GPSM2-GNAI complex, thereby organizing and lengthening cilia in sensory hair cells [91]. The cilium complex is an important structure in auditory cells. It consists of a series of proteins that form bundles of cilia; these structures play a key role in the perception of sound and other mechanical stimuli. It was found that the Whirlin-Myo15-Eps8 complex was able to form cohesions at the tips of the cilia by phase separation [92], which is essential for the regulation of the structure and function of the stereocilia complex. The cilium complex is mainly composed of proteins, including connecting proteins, supporting proteins, and regulatory proteins, which interact to form the structural basis of the cilium bundle.

Key components of the cilium bundle include, but are not limited to, Cdh23, Pcdh15, Myo7a, Espin, Harmonin, Sans, and others [69]. The molecular structure and conformational organization of Pcdh15 and its associated complexes within cilia have been extensively investigated by cryo-electron tomography [71]. The findings of the study provide a direct observation of the location and interaction of Pcdh15 in cilia. These proteins form a complex network of connections in ciliary bundles, which not only maintains their morphology and structural stability of cilia, but also regulates their functions [67]. LHFPL5 is required for the incorporation of PCDH15 into the MET complex, enabling its co-localization into the stereocilia of hair cells [20]. While the structure of the ciliary complex is highly ordered and complex, connexins form connections between ciliary bundles, support proteins provide mechanical support for ciliary bundles, and regulatory proteins regulate the activity and function of ciliary bundles [35,93]. Receptor cilia are cochlear cilia that form connections between cilia bundles. Receptor cilia, also known as stereocilia, are hair-like projections on the surface of cochlear hair cells that are involved in the process of mechanotransformation—the conversion of sound waves into electrical signals. Studies have shown that CIB2 and CIB3 are important for stereocilia maintenance and MET in mouse submarine hair cells [17]. The importance of BAIAP2L2 in the auditory system has been emphasized by a study showing that it is essential for the structural integrity and function of receptive cilia [94]. In addition, RAB11A has been shown to play a key role in the mammalian cochlea, where it is crucial for the development and structural integrity of auditory and balance hairs in auditory hair cells. Deletion or mutation of *Rab11a* leads to morphological abnormalities or even complete loss of auditory and balance hairs in the auditory hair cells, ultimately resulting in hearing and balance impairments [95].

Studies on the cilium complex in zebrafish have focused on exploring its composition, structure, and function to gain insight into the mechanisms of auditory perception. The lateral line system consists of neural hair cells that have morphologically mirror-symmetrically arranged bundles of hair cells that mechanically respond to forces in two opposite directions. It has been found that *tmc2b* has a differential role in mechanotransduction within the zebrafish lateral line system [96]. In *tmie* mutant zebrafish, the GFP-tagged MET channel subunit, *tmc1* and *tmc2b*, failed to localize to the stereociliary bundles. In contrast, overexpression of *tmie* significantly enhanced the localization of both *tmc1*-GFP and *tmc2b*-GFP to the cilia, indicating that Tmie is required for proper trafficking or anchoring of these MET channel components in sensory hair cells [36].

Through gene knockout, fluorescent labeling, and other techniques, the researchers found that the cilia complex in zebrafish is mainly composed of connecting proteins such as CDH23 and PCDH15, as well as supporting proteins like MYO7A and ESPIN. These proteins form complex structures in zebrafish cilia bundles that are involved in auditory perception processes [31,97]. In addition, a mutation in the structural domain of the non-USH protein glutathione-containing cysteine-rich 1 (GRXCR1) has been shown to cause non-syndromic sensorineural deafness. Another study constructed two *grxcr1* zebrafish mutants. It was found that *grxcr1* mutant zebrafish had thinner hair bundles, and further analysis showed that glutathionylation promotes the interaction between Ush1c and Ush1ga, while *Grxcr1* regulates mechanoreceptor development by inhibiting these interactions without disrupting the assembly of Ush1c-*Cdh23-Myo7a* complexes [98]. These findings collectively demonstrate that the morphogenesis of mechanoreceptors and proper mechanoelectrical transduction in sensory hair cells during zebrafish development critically depend on the integrity of ciliary bundles.

Previously, through electron microscopy, immunostaining, and other techniques, researchers observed the structure of cilium complexes in zebrafish auditory cells and found that they are formed by a complex network of multiple proteins interacting with each other. These structures support the morphology and stability of ciliary bundles while participating in the mechanisms of auditory perception [26,57]. Immunolabelling of zebrafish hair cells and studies using transgenic zebrafish expressing Fascin 2b-GFP, a filamentous actin bundling protein found in retinal photoreceptors, have shown that Fascin 2b is exclusively localized to stereocilia [99].

In conclusion, zebrafish, as a model organism, have unique advantages in auditory research, and the study of its stereocilia complex can help to reveal the mechanism of auditory perception and provide an important reference for the study of human auditory diseases.

## 9. Mutations of the Stereocilia Complex Associated with the Auditory System in Zebrafish

Abnormalities in the stereocilia complex have also been progressively revealed in relation to hearing impairment. TMC1 and TMC2, which are required for mechanotransduction in mouse inner ear hair cells, localize to the site of mechanotransduction of mouse hair cell stereocilia [100]. mTORC2, a cell-signaling complex involved in regulating cell proliferation, survival, and metabolism, has been implicated in sensory hair cell function. To investigate its role, hair cells (HCs)-specific Rictor knockout (HC-RicKO) mice were generated. Ultrastructural analysis revealed that short or absent hair cell cilia in external HCs caused HC-RicKO mice to exhibit early-onset, progressive, and profound hearing loss [101]. Moreover, mutations in genes involved in the ciliary complex also contribute to hearing deficits. For example, simultaneous knockout of *Cib2* and *Cib3* in mice disrupts the maintenance of stereocilia in all vestibular hair cells (VHCs), leading to severe balance deficits [17]. Findings from vestibular hair cells contribute to understanding the shared molecular machinery underlying both balance and hearing.

Polycystic kidney and liver disease 1-like protein (PKHD1L1), a large and predominantly extracellular protein, has also been identified as a critical component in the stereocilia complex. Sequential immunogold scanning electron microscopy demonstrated that PKHD1L1 is expressed in the apical portion of the cilia, particularly in the high-frequency region of the cochlea. In PKHD1L1-deficient mice, the upper portion of the stereocilia complex lacked a surface coating, while the lower portion remained unaffected. These mice exhibited progressive hearing loss, implicating PKHD1L1 in auditory function [102]. In addition, Gpsm2-Gαi complex has been shown to promote the coalescence of the first row of specific apical complexes, providing new insights into the identity of the arrangement of the highest hair cell cilia and offering possible clues to the etiology of hearing loss in patients with Chudley-McCullough syndrome [103]. Furthermore, the Dia1 protein, part of the hyaluronan-forming protein-related family Dia1, plays an important role in the development and maintenance of apical-joint complexes (AJCs) and cilia between HCs and supporting cells (SCs), ensuring cochlear and HC integrity [104]. The potential susceptibility of HCs may be the cause of progressive hearing loss in patients with DFNA1, providing a potential therapeutic target for preventing HC degeneration and DFNA1-associated progressive hearing loss.

In summary, abnormalities of the stereocilia complex are closely linked to hearing disorders and are crucial for understanding the regulatory mechanisms of auditory function and for treating these disorders. Future studies should further explore the mechanism of the stereocilia complex in auditory function and find new therapeutic targets to provide more effective methods for the treatment and prevention of hearing disorders.

## 10. Conclusions and Future Perspectives

Zebrafish have emerged as a powerful model for studying the molecular mechanisms underlying auditory and vestibular function. Their genetic tractability, transparent embryos, and conserved inner ear structures provide unique advantages for dissecting the pathogenesis of key components such as MET channels, tip link proteins, and the stereocilia complex. Zebrafish models enable real-time imaging and functional analysis of hair cell development, and allow for precise gene editing to investigate the roles of individual genes in auditory physiology.

Research using zebrafish has advanced our understanding of MET channels as critical mediators of mechanical-to-electrical signal conversion, uncovering regulatory factors that govern their assembly, trafficking, and function. tip link proteins such as Cdh23 and Pcdh15 have been shown to be essential for force transmission and hair cell connectivity, with zebrafish mutants mimicking human deafness phenotypes and revealing new insights into their spatial and temporal dynamics. Studies on the stereocilia complex have begun to identify the structural components and interactions that support its architecture, though many aspects of its assembly and maintenance remain to be clarified. Recent work has also highlighted the potential contribution of subtype-specific gene expression to sensory specialization. For example, Tmc subunits, including *tmc2b*, have been implicated in frequency sensitivity within the zebrafish utricle, suggesting a role in vestibular tuning [105]. Likewise, disruption of synaptojanin 1 (*synj1*) leads to temporal vestibular deficits, emphasizing the importance of precise gene expression patterns in sensory processing [106]. While these findings point to possible differences in how auditory and vestibular systems respond to altered expression of key genes, direct comparative studies—particularly for genes such as *tmc2b* and *lhfpl5b*—remain limited and represent an important avenue for future research.

Beyond mechanistic insights, zebrafish have been widely applied to the modeling of auditory and vestibular disorders. They facilitate gene discovery, pathophysiological studies, and high-throughput drug screening. Their regenerative capacity also offers a unique opportunity to explore hair cell repair and stem cell-based therapies. Furthermore, zebrafish are increasingly used to study gene–environment interactions, supporting investigations into how environmental stressors and genetic susceptibility converge in hearing loss.

Despite these strengths, limitations persist. The zebrafish auditory system is anatomically simpler than that of mammals, which may constrain its ability to fully recapitulate certain aspects of human disease. Functional assays for auditory perception remain technically challenging, and current genetic tools, while powerful, still face efficiency and stability issues. Environmental variables such as temperature and water quality can also impact reproducibility. Moreover, real-time imaging of inner ear activity is limited by current optical resolution and tissue depth constraints.

In summary, zebrafish represent an essential platform for advancing auditory biology. Continued integration of genetic, proteomic, and imaging technologies—alongside cross-species validation—will be critical for resolving remaining questions about the regulation of MET channels, tip link interactions, and stereocilia architecture. Ultimately, zebrafish studies will contribute to the development of precise diagnostic and therapeutic strategies for auditory and vestibular disorders.

## 11. Future Directions

Key open questions in zebrafish auditory biology include: (i) the precise molecular mechanisms of MET channel gating and modulation; (ii) the dynamics and regulation of tip link regeneration, and how zebrafish-specific mechanical properties compare to mammals; (iii) the functional significance of subtype-specific expression patterns of MET-associated genes in auditory versus vestibular hair cells; and (iv) the cellular and molecular pathways enabling regeneration after hair cell loss caused by mutations that lead to permanent deafness in mammals. Priority areas for future research include high-resolution structural studies of the MET complex, long-term tracking of regeneration after targeted genetic disruptions, and creating scalable in vivo assays to speed up drug discovery for hearing restoration.

## Figures and Tables

**Figure 1 ijms-26-08480-f001:**
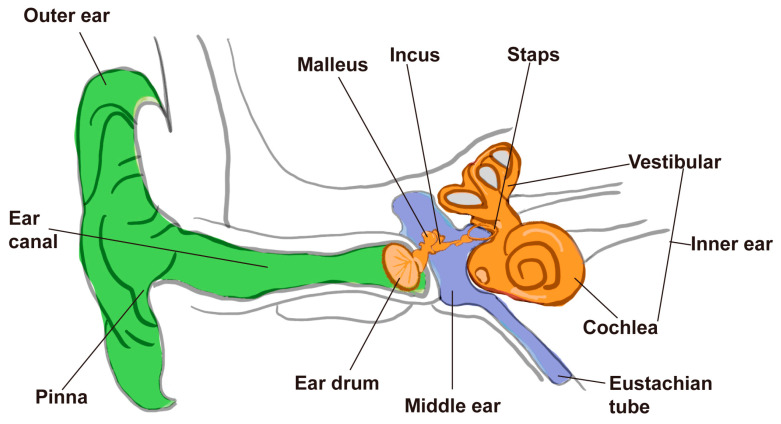
Anatomical structures of the mammalian outer, middle, and inner ear.

**Figure 2 ijms-26-08480-f002:**
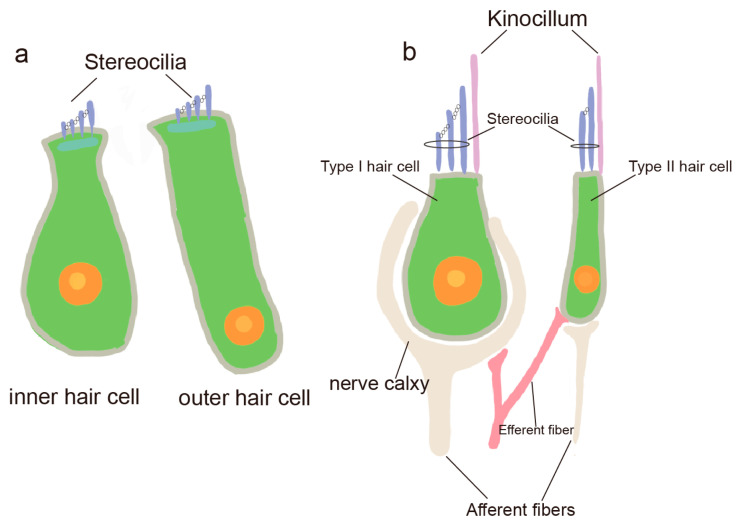
Types of IHCs and vestibular hair cells. (**a**) Cochlear inner hair cells (IHCs) and outer hair cells (OHCs) in the organ of Corti; (**b**) Type I and type II vestibular hair cells.

**Figure 3 ijms-26-08480-f003:**
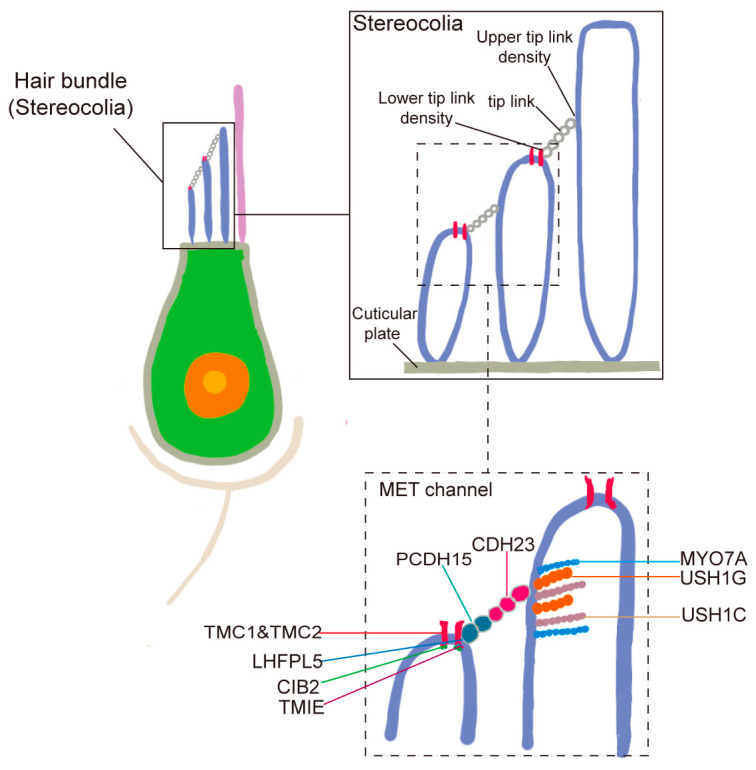
Complexes of hair cell stereocilia, tip link, and MET channels.

**Figure 4 ijms-26-08480-f004:**
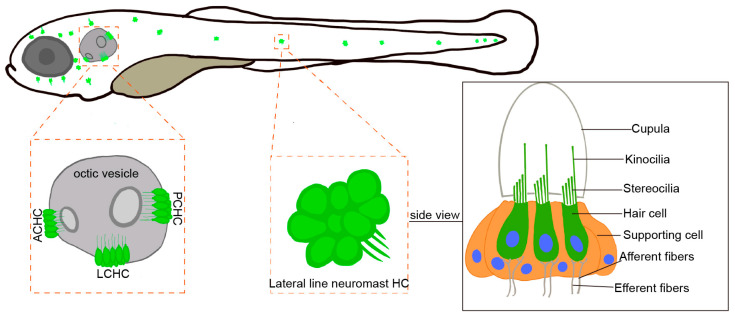
Zebrafish larva hair cells: anterior crista hair cell (ACHC), lateral crista hair cell (LCHC), and posterior crista hair cell (PCHC).

**Figure 5 ijms-26-08480-f005:**
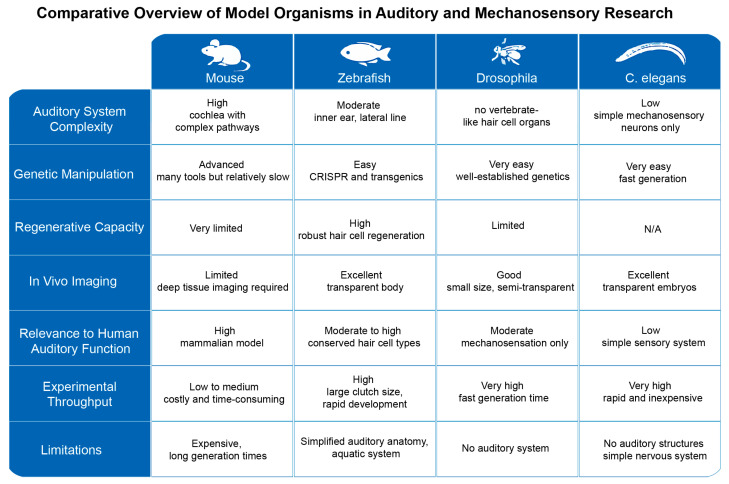
Comparative Overview of Model Organisms: Auditory and Mechanosensory Research.

## Data Availability

Not applicable.

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
