# Peer review of "Research Progress on MET, Tip Link, and Stereocilia Complex with Special Reference to Zebrafish"

_ijms, 2025, doi:10.3390/ijms26178480_

Round 1
Reviewer 1 Report
Comments and Suggestions for Authors
This review comprehensively summarizes recent advances in MET, tip-link, and stereocilia research using zebrafish models. The manuscript is well-structured, timely, and highlights the unique advantages of zebrafish for auditory studies. However, two key areas require strengthening: (1) critical discussion of zebrafish model limitations, and (2) visual synthesis of cross-model comparisons.
Major Comments
1. Inadequate Discussion of Zebrafish Model Limitations
While Section 2 (Zebrafish is an Important Model Organism for Studying Hair Cell Function) thoroughly details the strengths of zebrafish, it lacks a balanced critique of its limitations for auditory research.
2. Need for a Comparative Schematic Figure
The text compares zebrafish with mice, Drosophila, and C. elegans but lacks visual synthesis. A figure contrasting models would significantly enhance accessibility.
Author Response
Reviewer 1
Comment 1: Inadequate Discussion of Zebrafish Model Limitations
While Section 2 (Zebrafish is an Important Model Organism for Studying Hair Cell Function) thoroughly details the strengths of zebrafish, it lacks a balanced critique of its limitations for auditory research.
Response1:
We thank the reviewer for this constructive suggestion. In the revised manuscript, we have expanded Section 2 to include a detailed discussion of the limitations of the zebrafish model for auditory research. Specifically, we now highlight the anatomical simplicity of the zebrafish auditory system compared to mammals, the challenges in functional auditory assays, and the technical constraints involved in imaging deep inner ear structures. These updates provide a more balanced and comprehensive view of the model’s strengths and weaknesses (Lines 187-195).
Comment 2: Need for a Comparative Schematic Figure
The text compares zebrafish with mice, Drosophila, and C. elegans but lacks visual synthesis. A figure contrasting models would significantly enhance accessibility.
Response2:
We appreciate this valuable suggestion. In the revised manuscript, we have added a new schematic figure (Figure 5) that visually summarizes the key similarities and differences between zebrafish, mice, Drosophila, and C. elegans in auditory and mechanosensory research. This figure integrates anatomical, genetic, and experimental advantages/limitations into a comparative layout, which we believe will improve accessibility and facilitate cross-model understanding for readers (Lines 196-202).
Reviewer 2 Report
Comments and Suggestions for Authors
This review of MET in zebrafish nicely brings together a considerable amount of good science by many investigators. A major flaw of this manuscript is the figures. Figures 1, 2 and 3 are amateurish as regards details although admittedly the figures are artistic and colorful. Either provide anatomically detailed drawings of the outer ear, middle ear and inner ear with relevant labels of the structures, for example, or delete these figures. As an example, figures 2 and 3 lacks details of stereocilia structure and are scientifically worthless in contrast to the nice quality of the text of this review.
The history of the discovery of mammalian TMC1 as the MET channel component did not begin with the report described in reference 11 but rather with human genetics of deafness DFNB7/11. Tip links is another example that began with the discovery of variants underlying Usher syndrome type 1D and nonsyndromic deafness DFNB12 (CDH23), and Usher syndrome type 1F and DFNB23 (PCDH15). If this review is going to be credible and scholarly, a proper history and appropriate references are needed, which should not add more than a few sentences to the text and a few more references.
Minor edits
Genus and species names are italicized such as C elegans (line 68)
Lines 158-160, The authors state: “It has been found that zebrafish Tmc1 and Tmc2a can interact with the CD1 or CD3 cytoplasmic structural domain isoforms of Pcdh15a and affect the stability of cilia structure[28]” Please explain what is meant by CD1 and CD3 and properly reference.
Figure 4, This figure is also not professionally illustrated and lacks minimal details of the structural features of a zebra fish neuromast.
Line 182-184, The authors state “These channels are located at the tips of IHCs and sense and convert signals in response to sound-induced mechanical stimuli.” This statement is not true. The authors drew correctly the location of the MET channels near the tips of shorter row stereocilia in Figure 3 and so they must know that the MET channel is absent from the tallest stereocilia row.
Consider not capitalizing the “t” in tip link except when “Tip” begins a sentence.
Author Response
Reviewer 2
Comment 1:
This review of MET in zebrafish nicely brings together a considerable amount of good science by many investigators. A major flaw of this manuscript is the figures. Figures 1, 2 and 3 are amateurish as regards details although admittedly the figures are artistic and colorful. Either provide anatomically detailed drawings of the outer ear, middle ear and inner ear with relevant labels of the structures, for example, or delete these figures. As an example, figures 2 and 3 lacks details of stereocilia structure and are scientifically worthless in contrast to the nice quality of the text of this review.
Response 1:
We thank the reviewer for this constructive feedback. Figure 1 has been revised with anatomically detailed outer, middle, and inner ear structures. Figures 2 and 3 now include additional stereocilia details.
Comment 2:
The history of the discovery of mammalian TMC1 as the MET channel component did not begin with the report described in reference 11 but rather with human genetics of deafness DFNB7/11. Tip links is another example that began with the discovery of variants underlying Usher syndrome type 1D and nonsyndromic deafness DFNB12 (CDH23), and Usher syndrome type 1F and DFNB23 (PCDH15). If this review is going to be credible and scholarly, a proper history and appropriate references are needed, which should not add more than a few sentences to the text and a few more references.
Response 2:
We thank the reviewer for pointing out this error. We agree with this comment and have revised the introduction to include a more complete historical background:
Added the historical identification of CDH23 and PCDH15 variants in Usher syndrome and nonsyndromic deafness (Lines 66-71).
Added the initial discovery of DFNB7/11 and its link to TMC1 in human genetic studies (Lines 74-77).
We have also included appropriate references [New references #10-13] to support these updates.
Comment 3:
Genus and species names should be italicized (e.g., C. elegans, line 68).
Response 3:
We thank the reviewer for pointing out this issue. We have corrected genus and species names throughout the manuscript. For example, C. elegans is now italicized (Line 80).
Comment 4:
Lines 158-160, The authors state: “It has been found that zebrafish Tmc1 and Tmc2a can interact with the CD1 or CD3 cytoplasmic structural domain isoforms of Pcdh15a and affect the stability of cilia structure[28]” Please explain what is meant by CD1 and CD3 and properly reference.
Response 4:
We have clarified that “Zebrafish Tmc1 and Tmc2a have been shown to interact with distinct cytoplasmic domain isoforms of Pcdh15a, referred to as CD1 and CD3, which differ in their C-terminal sequences. These isoforms differentially affect the stability and assembly of the mechano-transduction complex within stereocilia.” The sentence has been revised for clarity (Lines 171-175) and the reference updated to include the original source [Updated Reference #31].
Comment 5:
Figure 4, This figure is also not professionally illustrated and lacks minimal details of the structural features of a zebra fish neuromast.
Response 5:
We have replaced Figure 4 with a new one, an anatomically accurate depiction of a zebrafish neuromast, including detailed features such as kinocilia, stereocilia rows, supporting cells, and afferent/efferent innervation (Figure 4).
Comment 6:
Line 182-184, The authors state “These channels are located at the tips of IHCs and sense and convert signals in response to sound-induced mechanical stimuli.” This statement is not true. The authors drew correctly the location of the MET channels near the tips of shorter row stereocilia in Figure 3 and so they must know that the MET channel is absent from the tallest stereocilia row.
Response 6:
We thank the reviewer for this correction. The sentence has been revised to accurately state:
“These channels are located near the tips of the shorter row stereocilia and convert sound-induced mechanical stimuli into electrical signals.” (Lines 214-216).
Comment 7:
Consider not capitalizing the “t” in tip link except when “Tip” begins a sentence.
Response 7:
We have revised the manuscript to ensure “tip link” is only capitalized at the beginning of a sentence.
Reviewer 3 Report
Comments and Suggestions for Authors
This is a well-written and thorough review that effectively summarizes current understanding of mechano-electrical transduction (MET) channels, Tip-link proteins, and the stereocilia complex, with a particular focus on zebrafish as a model system. The sections addressing genetic mutations and their phenotypic consequences in zebrafish are particularly valuable for linking fundamental research to human auditory disorders. Figures are generally appropriate for illustrating anatomical and molecular relationships.
Specific Comments
- Some sections are relatively detailed; such as those on non-auditory mechanosensory systems. These could benefit from a more explicit connection back to the central theme of auditory function to help maintain narrative cohesion.
- Ensure that all figures are cited and discussed within the text, with adequate contextual explanation. For readers less familiar with the field, expanding figure captions to include more explanatory content would be beneficial.
- The conclusion could be expanded to more explicitly outline key open questions and research priorities for zebrafish auditory biology, particularly with regard to unresolved aspects of MET channel gating and Tip-link regeneration.
- The review mentions that zebrafish Tip-link proteins may exhibit unique mechanosensory properties (slip–ideal–slip behavior). It would be valuable for the authors to elaborate on how this differs from mammalian Tip-links and to discuss potential functional implications.
- Several zebrafish genes (e.g., tmc2b, lhfpl5b) are described as having subtype-specific or reduced expression in certain hair cell populations. The authors could discuss how these expression patterns might contribute to differences in auditory versus vestibular sensitivity.
- The discussion of regenerative capacity in zebrafish hair cells would be strengthened by noting whether there are examples in which MET or Tip-link mutations causing permanent deafness in mammals have been functionally recovered in zebrafish through regeneration.
- The section on pharmacological screening for MET channel modulators could benefit from more detail on how zebrafish behavioral or electrophysiological assays are adapted for high-throughput drug discovery.
Author Response
Reviewer 3
Comment 1:
Some sections are relatively detailed; such as those on non-auditory mechanosensory systems. These could benefit from a more explicit connection back to the central theme of auditory function to help maintain narrative cohesion.
Response 1:
We agree with this comment. In the revised manuscript, we have added bridging sentences at the end of each relevant subsection to explicitly link non-auditory mechanosensory systems to auditory hair cell biology. For example, in the section on lateral line mechanosensory systems (lines 411-413) and vestibular hair cells (lines 643-644), we now emphasize how findings from these systems provide insights into hair cell polarity and mechanotransduction relevant to auditory function.
Comment 2:
Ensure that all figures are cited and discussed within the text, with adequate contextual explanation. For readers less familiar with the field, expanding figure captions to include more explanatory content would be beneficial.
Response 2:
We have thoroughly checked the manuscript to ensure every figure is explicitly cited and discussed in the main text. Additionally, we have expanded figure captions to provide more detailed descriptions, including key anatomical and molecular features for readers less familiar with the field (Figures 1–4, lines 58-60).
Comment 3:
The conclusion could be expanded to more explicitly outline key open questions and research priorities for zebrafish auditory biology, particularly with regard to unresolved aspects of MET channel gating and Tip-link regeneration.
Response 3:
We thank the reviewer for this valuable suggestion. We have expanded the conclusion to explicitly outline key open questions and research priorities in zebrafish auditory biology. The new content (Lines 712-722) summarizes unresolved mechanisms of MET channel gating, tip-link regeneration, and gene expression heterogeneity, and highlights priority areas such as high-resolution structural studies, regeneration tracking, and scalable in vivo screening.
Comment 4:
The review mentions that zebrafish Tip-link proteins may exhibit unique mechanosensory properties (slip–ideal–slip behavior). It would be valuable for the authors to elaborate on how this differs from mammalian Tip-links and to discuss potential functional implications.
Response 4:
We thank the reviewer for this insightful comment. While slip-ideal-slip behavior in tip-link complexes has been demonstrated in recent work (e.g., Arora et al., Nat Commun 2024)(Reference 83), there is currently no published evidence demonstrating this phenomenon for zebrafish tip-link proteins.
Comment 5:
Several zebrafish genes (e.g., tmc2b, lhfpl5b) are described as having subtype-specific or reduced expression in certain hair cell populations. The authors could discuss how these expression patterns might contribute to differences in auditory versus vestibular sensitivity.
Response 5:
We thank the reviewer for this insightful comment. Recent studies provide some relevant evidence. For example, Tmc subtype, including tmc2b, has been implicated in frequency sensitivity within the zebrafish utricle, suggesting a potential role in tuning vestibular function [Updated Reference #105] (Lines 682-685). Additionally, subtype-specific gene disruption, such as in synaptojanin 1 (synj1) mutants, results in temporal vestibular deficits, further highlighting the importance of precise gene expression patterns in sensory processing [Updated Reference #106] (Lines 686-687). However, direct comparisons of how reduced or differential expression of genes such as tmc2b and lhfpl5b affects auditory versus vestibular sensitivity remain limited. We have acknowledged this as an important open question and future research priority in the revised manuscript, consistent with the key open questions mentioned elsewhere in the paper.
Comment 6:
The discussion of regenerative capacity in zebrafish hair cells would be strengthened by noting whether there are examples in which MET or Tip-link mutations causing permanent deafness in mammals have been functionally recovered in zebrafish through regeneration.
Response 6:
We thank the reviewer for this valuable suggestion. Zebrafish possess a remarkable capacity for hair cell regeneration that allows recovery of mechanosensory function after damage. Although direct evidence showing recovery of specific MET or tip-link mutations—which cause permanent deafness in mammals—via regeneration in zebrafish is still lacking, recent studies have begun to reveal the molecular programs that enable such regenerative capacity. For example, stem and progenitor cell proliferation in the zebrafish inner ear can be regulated by cell–type–specific cyclin D genes, indicating that intrinsic control of regenerative cell cycles may provide a basis for restoring function after genetic lesions [Updated Reference #81]. We have revised the discussion to highlight these findings and to emphasize the potential of zebrafish as a model for exploring regenerative therapies for genetic hearing loss (Lines 503-509).
Comment 7:
The section on pharmacological screening for MET channel modulators could benefit from more detail on how zebrafish behavioral or electrophysiological assays are adapted for high-throughput drug discovery.
Response 7:
We have expanded the manuscript to provide more detail on how zebrafish assays are adapted for high-throughput pharmacological screening of MET channel modulators. This enhanced description can be found in the revised manuscript (Lines 277-283).
Round 2
Reviewer 2 Report
Comments and Suggestions for Authors
The manuscript is improved, the English is fine and the figures are also much improved. No concerns with the revised manuscript. "link" in the title should be capitalized.